# Effects of Taekwondo intervention on balance ability: A meta-analysis and systematic review

Zhengfa Han[1], Hanyu Ju[2]*

1 School of Physical Education, Guangdong University of Education, Guangzhou, China, 2 School of Physical Education, University of Sanya, Sanya, China

* juhanyu123@163.com

**Data Availability Statement:** All relevant data are within the paper and its Supporting information files.

## Abstract

### Objective

This study investigated the effects of Taekwondo interventions on balance ability through meta-analysis and systematic review. Additionally, an optimal intervention protocol was proposed based on subgroup analysis of single-leg stance outcomes to enhance the overall effect.

### Methods

Systematic searches were conducted in Chinese (CNKI, WANFANG DATA), English (Web of Science, PubMed), and Korean (KISS, RISS, DBPIA) databases to identify randomized controlled trials (RCTs), pre-post studies, and cross-sectional studies (CSS) evaluating Taekwondo's impact on balance. A total of 21 studies encompassing 972 participants were included. The risk of bias in the RCTs was assessed using the Cochrane Risk of Bias Tool version 2 (RoB 2.0), and the methodological quality of pre-post and cross-sectional studies was evaluated with the Methodology Index for Non-Randomized Studies (MINORS) criteria. Quantitative analyses focused on measures such as single-leg stance with eyes closed, marching in place with eyes closed, and assessments using the Win pod device. Qualitative analyses addressed other assessment methods.

### Results

The overall risk of bias results for the included RCTs indicated that four studies were classified as high risk, while the remaining studies demonstrated some concerns regarding risk. The methodological quality of the included non-RCTs was assessed as moderate. Furthermore, the quantitative outcomes indicated that Taekwondo interventions significantly enhanced both static (single-leg stance with eyes closed; $ES = 0.862$, $p<0.001$) and dynamic (marching in place with eyes closed; $ES = 0.296$, $p = 0.036$) balance abilities. However, analysis using the Win pod device for static balance showed no significant improvement ($p>0.05$). Subgroup analysis for the single-leg stance with eyes closed demonstrated the most substantial effects in adults ($ES = 1.191$, $p = 0.001$) and females ($ES = 0.786$, $p = 0.005$). The most effective Taekwondo intervention featured a duration of 12 weeks ($ES = 1.375$, $p = 0.002$), a frequency of once per week ($ES = 1.406$, $p = 0.003$), and a times length

**Funding:** This study complies with the current laws of the country/region where it was conducted, and there are no conflicts of interest. The study was funded by the Talent Introduction Project of Sanya University (USYRC24-07). The funders had no role in study design, data collection and analysis, decision to publish, or preparation of the manuscript.

**Competing interests:** The authors have declared that no competing interests exist.

of 60–70 minutes ($ES$ = 1.028, $p$<0.001). Qualitative assessments supported these findings, indicating overall beneficial impacts on balance from Taekwondo training across various populations and evaluation methods.

## Conclusions

In conclusion, Taekwondo interventions are effective for enhancing static and dynamic balance abilities, especially in adult females. It is recommended to follow a training protocol of 60–70 minutes per times, once weekly, for 12 weeks to optimize static balance improvements.

## Introduction

Balance is a critical skill necessary for maintaining posture and preventing falls during standing, walking, or other physical activities [1]. Balance can be classified into two types: static balance, which pertains to stability when holding a particular posture [2], and dynamic balance, which involves adjusting one's posture during motion [3]. This ability is essential for daily activities and plays a significant role in preventing sports-related injuries [4, 5]. Additionally, proficient balance is fundamental to success in numerous sports disciplines [6]. As a result, enhancing balance ability is a focal point of academic research [4, 7–9]. Despite ongoing exploration of new methods to improve balance, these techniques are primarily based on physical exercises. Consistent physical activity is a proven method to enhance balance ability [10–12].

As an official Olympic sport, Taekwondo enjoys immense popularity worldwide [13]. Over time, it has transitioned from a combat sport limited to athletes to a comprehensive discipline accessible to practitioners from diverse backgrounds [14]. Some studies have indicated that Taekwondo can positively impact balance ability [15–17]. However, while significant enhancements in static balance have been reported, similar advancements in dynamic balance remain unobserved [17, 18]. Additionally, research utilizing closed-eye one-leg standing tests to evaluate the effects of Taekwondo interventions on balance ability indicated notable improvements in balance after four weeks of intervention, with no further changes observed beyond eight weeks. This study also employed the Berg Balance Scale to assess the intervention's impact, but the results showed no significant enhancements in balance ability [19]. Furthermore, a study employing Win pod devices to track center of pressure trajectories revealed that an 18-week Taekwondo intervention significantly improved only two indicators (Area and X deviation) in overweight children, with no significant results for other metrics [20]. The current body of research presents inconsistent findings on the effectiveness of Taekwondo interventions in enhancing balance ability. To date, no systematic reviews or meta-analyses have been conducted to synthesize the impacts of Taekwondo on balance. Thus, this study utilizes meta-analysis and systematic review methodologies to quantitatively and qualitatively aggregate existing research on Taekwondo's effects on balance ability. The findings will provide evidence-based recommendations for exercise interventions aimed at improving balance.

## Methodology

This meta-analysis and systematic review was conducted following the PRISMA (Preferred Reporting Items for Systematic Reviews and Meta-Analyses) 2020 Checklist and has been registered in PROSPERO (registration number: CRD42024555656).

## Search strategy

To ensure comprehensive retrieval of pertinent literature, databases in three languages—Chinese (CNKI, WANFANG DATA), English (Web of Science, PubMed), and Korean (KISS, RISS, DBPIA)—were systematically searched. The study began in June 2024, and the literature search was conducted in July 2024, encompassing articles published up to and including July 2024. The search strategy employed was "keyword 1 + keyword 2," using the full terms for each keyword. "Taekwondo" was designated as keyword 1. Given the various expressions related to "balance ability" in the Chinese context, such as "Balance" and "Balance control Ability," and the differentiation into "static balance ability" and "dynamic balance ability," keyword 2 was defined as "Balance." This approach aimed to capture a broad spectrum of relevant studies. During the search, documents other than dissertations and journal articles, such as conference papers, newspapers, and patents, were excluded. A total of 593 articles were ultimately retrieved, including 261 in Chinese, 30 in English, and 302 in Korean.

## Inclusion and exclusion criteria

This meta-analysis and systematic review applied the PICOS (Population, Intervention, Comparison, Outcome, and Study design) framework to define inclusion and exclusion criteria. Participants (P): The study included healthy males and females of all ages, excluding individuals with movement disorders, developmental disabilities, and athletes. Interventions (I): The analysis incorporated all forms of Taekwondo interventions, such as poomsae, kick, and Taekwondo gymnastics. Studies that included auxiliary interventions like medication or dietary changes were excluded. Control group (C): The control group consisted of individuals who engaged in regular physical activities as part of their daily routines. Outcome measures (O): The primary outcome measured was balance ability. Study type (S): The study types included were randomized controlled trials (RCTs), before-and-after studies, and cross-sectional studies. Additionally, studies were excluded if they lacked extractable effect sizes from the reported data.

## Data extraction

Data extraction for this meta-analysis and systematic review was independently performed by two researchers. Upon completion of data extraction by both researchers, the collected data underwent cross-checking to ensure accuracy and consistency. Discrepancies encountered during this process were resolved through discussion among the research team. Literature screening was conducted using Zotero (version 6.0) software [21], which facilitated the organization and retrieval of relevant studies. The following information was systematically extracted from the selected articles: first author, published year, sample size, sex, age, measurement device or method, evaluation parameters, period, times per week, minutes per time, and exercise intensity.

## Methodological quality and publication bias assessment of included literature

For the included randomized controlled trials (RCTs), the risk of bias was assessed using the Cochrane Risk of Bias Tool version 2 (RoB 2.0) [22]. This tool classifies the risk of bias in studies into five domains: randomization process, deviations from intended interventions, missing outcome data, measurement of the outcome, and selection of the reported result. Each domain is evaluated and assigned one of three risk levels: low risk, some concerns, or high risk. Additionally, to assess publication bias, funnel plots and Egger's test were utilized within the Comprehensive Meta-Analysis software.

Given that the Cochrane Risk of Bias Tool version 2 (RoB 2.0) is tailored primarily for assessing bias in randomized controlled trials (RCTs), the methodological quality of included before-and-after studies and cross-sectional studies was assessed using the Methodology Index for Non-Randomized Studies (MINORS) items [23]. The MINORS tool includes 12 items, with the final four specifically designed for evaluating studies with control groups. Consequently, only the first eight items were employed to assess the methodological quality of the included before-and-after studies and cross-sectional studies. Each item in MINORS is scored on a scale where 0 indicates not reported, 1 signifies reported but inadequate, and 2 denotes reported and adequate. The maximum possible score for the first eight items is 16. Scores below 8 are generally considered indicative of low quality, scores between 8 and 12 suggest moderate quality, and scores above 13 are indicative of high quality [24].

## Statistical analysis

This meta-analysis and systematic review utilized the Risk of Bias 2.0 (RoB 2.0) tool to assess risk of bias, and Comprehensive Meta-Analysis software for conducting other data analyses. The standardized mean difference (SMD) using Hedges' g was selected as the final combined effect size measure because it corrects for the positive bias typically associated with Cohen's d [25]. According to Cohen's criteria, effect sizes were classified into three categories: 0.2 as small, 0.5 as moderate, and 0.8 as large [26]. The choice between a random-effects model and a fixed-effects model was based on the outcomes of heterogeneity tests, including $I^2$ statistics and p-values. Typically, a random-effects model is applied if $I^2$ is 50% or higher and p-value is less than 0.05; otherwise, a fixed-effects model is considered appropriate. It is important to note, however, that heterogeneity should not be inferred solely from $I^2$ and p-values, as underlying variations among studies also need to be taken into account [27]. Additionally, For all statistical analyses in this study, statistical significance was determined at a threshold of $p < 0.05$.

## Results

### Overview of included literature

The literature screening process is illustrated in Fig 1. Following this process, a total of 21 studies were selected for inclusion in this meta-analysis and systematic review. These studies included 13 conducted in Chinese [17, 18, 20, 28–37], 6 in Korean [19, 38–42], and 2 in English [15, 16]. The composition of the included studies encompassed 12 randomized controlled trials (RCTs) [15, 17–19, 30, 32, 33, 38–42], 5 before-and-after studies [16, 28, 35–37], and 4 cross-sectional studies [20, 29, 31, 34], totaling 972 participants. Detailed descriptions of these studies are provided in Tables 1 and 2. The participants' age varied widely, from 3 to 6 years [28] to 74 to 76 years [36], with intervention durations ranging from 6 weeks [38] to 15 months [16]. The frequency of interventions per week varied from once [17, 32, 35] to five times [19, 36], and minutes per times ranged from 40 minutes [19] to 120 minutes [17, 41]. Notably, only 7 studies reported the intensity of the interventions, all classified as moderate [18, 33, 35, 36, 40–42].

In assessing static balance ability, most studies used the "single-leg stance with eyes closed" method [15–19, 30, 32, 33, 35–37, 41]. Some studies employed specific equipment such as the "Force Plate" [39], "Balance Performance Monitor" [38], "Win pod" [20, 29, 34], and "Wii-Fit balance board" [31] to measure static balance. For dynamic balance ability, various methods were utilized, including "marching in place with eyes closed" [17, 18, 26], "walking on a balance beam" [18, 31], "Y balance test" [31, 41], "single-leg hop" [16], "timed up-and-go test (TUGT)" [15], "functional reach test (FRT)" [15], "Star Excursion Balance Test" [20, 29] and equipment

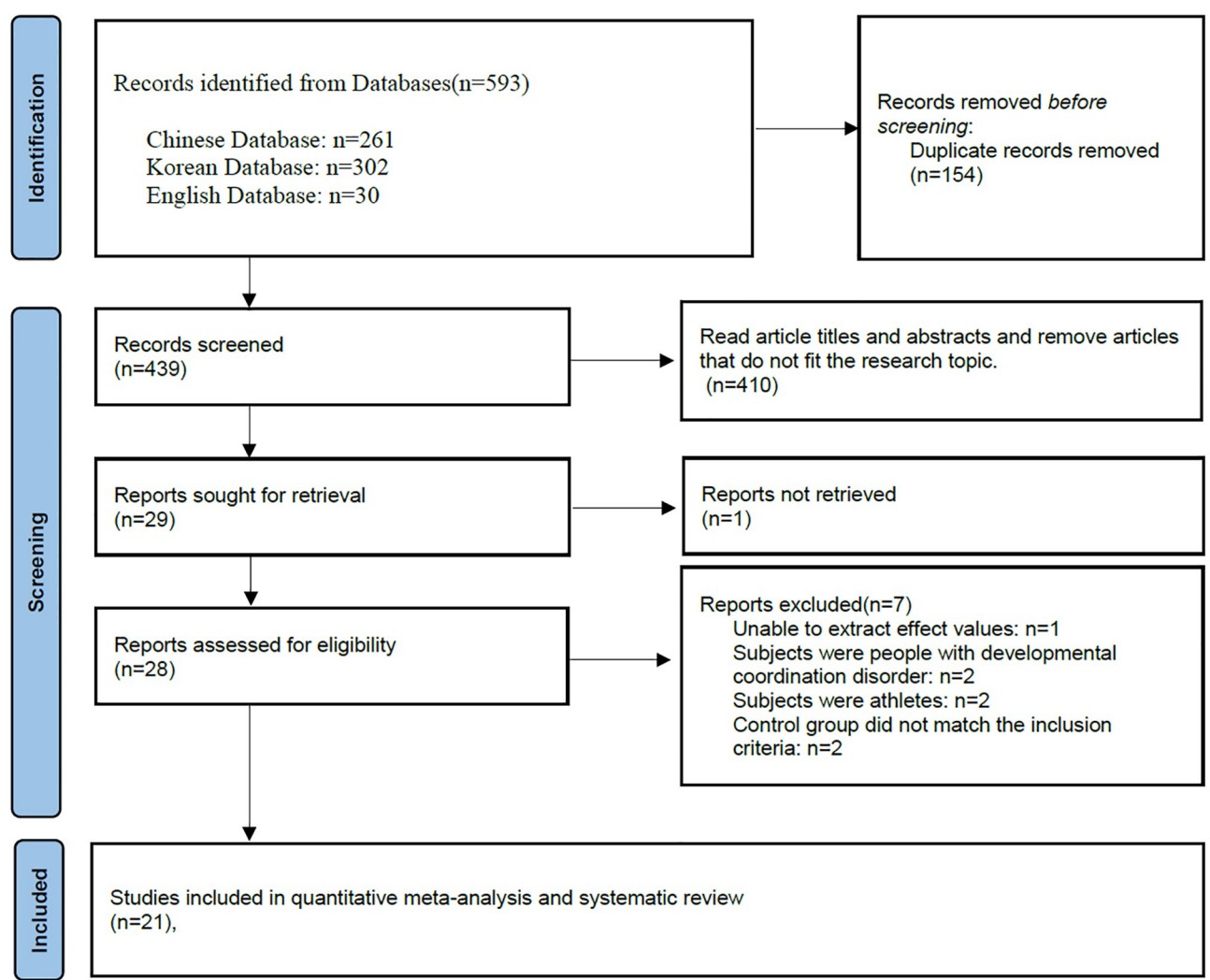

**Fig 1. PRISMA 2020 flow diagram.**

such as "GAITRite" [38] and "Good Balance System" [40, 42]. Additionally, the Berg Balance Scale was used in two studies for comprehensive balance assessment [19, 38].

## Methodological quality assessment of included literature

The methodological risk of bias for the 12 included RCTs is detailed in Figs 2 and 3. Of these, four studies exhibited a high risk in the randomization process [19, 38, 39, 41], while seven studies raised some concerns in this area [17, 18, 30, 32, 33, 40, 42], and the remaining study was assessed as low risk. Regarding overall bias, four studies were classified as high risk [19, 38, 39, 41], while the remaining studies showed some concerns.

The methodological quality of the five included before-and-after studies and four cross-sectional studies is detailed in Table 3. All studies were assessed as moderate quality, with scores exceeding 8. The methodological quality assessments indicated notable similarities across these studies. Specifically, none of the studies implemented blinding of outcome assessors nor conducted follow-up assessments, which resulted in scores of 0 for items 5–7. While all studies

**Table 1. Basic information of included RCTs and before-and-after studies.**

| The first author (Published year) | Sample size | | Sex | Age | Measurement device or method | Evaluation parameters | Intervention program | | | intensity |
|---|---|---|---|---|---|---|---|---|---|---|
| | Intervention | Control | | | | | Period | Times per week | Minutes per time | |
| Chang (2022) | 30 | | Unrestricted | 3–6 | **dynamic:** marching in place with eyes closed | **dynamic:** duration of marching in place (within a 20 cm radius circle) | 12week | 2 | 60min | |
| Cui (2022) | 20 | 20 | Unrestricted | 5–6 | **static:** single-leg stance with eyes closed **dynamic:** walking on a balance beam, marching in place with eyes closed | **static:** single-leg stance duration **dynamic:** balance beam walking duration, duration of marching in place (within a 20 cm radius circle) | 12 week | 2 | 45min | moderate intensity (HRavg120-140 bpm) |
| Zhang (2022) | 23 | 23 | Unrestricted | 4–6 | **static:** single-leg stance with eyes closed **dynamic:** marching in place with eyes closed | **static:** single-leg stance duration **dynamic:** duration of marching in place (within a 20 cm radius circle) | 16 week | 1 | 120min | |
| Huang (2020) | 30 | 30 | Unrestricted | 4–6 | **static:** single-leg stance with eyes closed | **static:** single-leg stance duration | 4 months | 1 | 90min | |
| Fu (2014) | 20 | 20 | Male, Female | 9 | **static:** single-leg stance with eyes closed | **static:** single-leg stance duration | 12 week | | | |
| Shao (2023) | 20 | | Unrestricted | 18–20 | **static:** single-leg stance with eyes closed | **static:** single-leg stance duration | 12 week | 1 | 90min | moderate intensity ((HRavg110-140 bpm)) |
| Yang (2014) | 36 38 | | Male, Female | 18–20 | **static:** single-leg stance with eyes closed | **static:** single-leg stance duration | 16 week | 2–3 | 90min | |
| Li (2020) | 50 | 50 | Unrestricted | 4–6 | **static:** single-leg stance with eyes closed | **static:** single-leg stance duration | 16 week | 2 | 45min | moderate intensity (HRmax 70%) |
| Wang (2018) | 225 | | Female | 18–20 | **static:** single-leg stance with eyes closed | **static:** single-leg stance duration | 15 week | 5 | 70min | moderate intensity ((HRavg120-140 bpm)) |
| Shin (2008) | 8 | 8 | Female | 68–71 | **static:** Force Plate | **static:** Cop Path Length, Cop Sway Area, Cop Velocity | 12 week | 3 | 60min | |
| Lee (2017) | 15 | 15 | Female | 19–21 | **static:** Single-Leg Stance with Eyes Closed dynamic: Y balance test | **static:** single-leg stance duration **dynamic:** absolute value of lower limb extension | 10 week | 2 | 120min | moderate intensity (RPE12-15) |
| Lee (2014) | 10 | 10 | Male | 12–13 | **dynamic:** Good Balance System | **dynamic:** anterior-posterior and medial-lateral sway distance, center of pressure displacement distance | 12 week | 3 | 60min | moderate intensity (HRmax 40–70%) |
| Jung (2014) | 10 | 10 | Unrestricted | 12–13 | **dynamic:** Good Balance System | **dynamic:** anterior-posterior and medial-lateral sway distance, center of pressure displacement distance | 12 week | 3 | 60min | moderate intensity (HRmax 40–70%) |

*(Continued)*

**Table 1.** (Continued)

| The first author (Published year) | Sample size | | Sex | Age | Measurement device or method | Evaluation parameters | Intervention program | | | intensity |
|---|---|---|---|---|---|---|---|---|---|---|
| | Intervention | Control | | | | | Period | Times per week | Minutes per time | |
| Kim (2008) | 17 | 9 | Unrestricted | 74–76 | **static:** Balance Performance Monitor **dynamic:** GAITRite **scale:** Berg Balance Scale | **static:** maximum sway velocity, sway distance, sway area **dynamic:** stride length, step length, foot support area scale: score | 6 week | 3 | 45min | |
| Yoon (2009) | 20 | 20 | Male | 12–13 | **static:** single-leg stance with eyes closed **scale:** Berg Balance Scale | **static:** single-leg stance duration **scale:** score | 8 week | 5 | 40min | |
| Pons Van Dijk (2013) | 17 | | Unrestricted | 41–70 | **static:** single-leg stance with eyes closed **dynamic:** single-leg hop | **static:** single-leg stance duration **dynamic:** single-leg hop distance | 15 months | 40* | 60min | |
| Kim (2024) | 25 | 23 | Female | 65–75 | **static:** single-leg stance with eyes closed **dynamic:** timed up-and-go test, functional reach test | **static:** single-leg stance duration dynamic: total duration, finger-to-toe distance | 12 week | 2 | 60min | |

*Blank: Not Reported; 40*: This study conducted a total of 40 training sessions over 15 months.

documented the criteria for determining statistical significance, the methods used to determine sample size were not reported, hence, item 8 received a score of 1. Additionally, one study was scored 1 for item 3 due to a low participant attendance rate, which necessitated an extended intervention period. In this particular study, several participants made up missed timess at locations outside the designated intervention site, while still adhering to the predetermined intervention protocol [16].

### Quantitative synthesis

This study categorized the intervention studies (RCTs and pre-post studies) and non-intervention studies (cross-sectional studies) for quantitative synthesis separately. Due to the diversity

**Table 2. Basic information of included cross-sectional studies.**

| The first author (Published year) | Sample size | | Sex | Age | Measurement Equipment or method | Evaluation parameters |
|---|---|---|---|---|---|---|
| | Intervention | Control | | | | |
| Pang (2016) | 30 | 30 | Unrestricted | 6–12 | **static:** Win pod **dynamic:** star excursion balance test | **Static:** Length, Area, Avg.v, X speed, Y speed, X dev., Y dev. **Dynamic:** Reach distances in six directions: left anterior, left posteromedial, left posterolateral, right anterior, right posteromedial, right posterolateral |
| Guo (2023) | 40 | 38 | Unrestricted | 4–5 | **static:** Wii-Fit balance board **dynamic:** walking on a balance beam, Y balance test | **Static:** Length, Area, Time **dynamic:** completion time, absolute value of lower limb extension, |
| Pang (2015) | 10 | 10 | Unrestricted | 7–8 | **static:** Win-pod | **static:** Length, Area, Avg.v, X speed, Y speed, X dev., Y dev. |
| Chen (2016) | 15/15 | 15/15 | Unrestricted | 6–8 9–12 | **static:** Win-pod **dynamic:** Star Excursion Balance Test | **static:** Length, Area, Avg.v, X speed, Y speed, X dev., Y dev. **dynamic:** absolute value of lower limb extension |

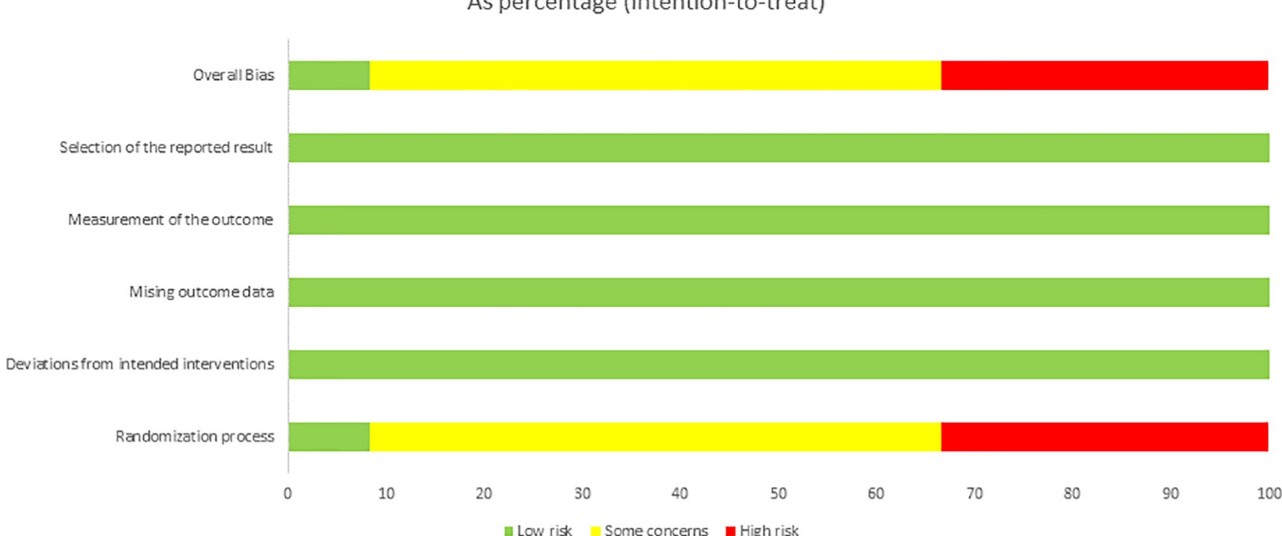

**Fig 2. Results of the RCTs risk of bias: Judgments about each risk of bias item for each included trail.**

in methods, tools, and metrics employed to assess balance ability, not all studies were suitable for inclusion in a meta-analysis. To maintain the representativeness of the meta-analysis results [43], only studies that used the same methods, tools, and metrics were included, provided that at least three such studies were available. Consequently, this research conducted meta-analyses using "closed-eye one-leg standing" and "Win pod" for assessing static balance ability, and "closed-eye marching in place" for dynamic balance ability. Studies that did not meet these criteria were subjected to qualitative analysis.

**Closed-eye one-leg standing.** *Main effect analysis.* Among the included studies, 12 intervention studies utilized the single-leg stance with eyes closed method to assess static balance. Two of these studies provided data separately for males and females, resulting in a total of 14 data sets (treated as 14 studies) for the meta-analysis. When data for both the left and right legs were available, only the data for the dominant leg (assumed to be the left leg) was included in the analysis. The results of the heterogeneity test and the meta-analysis for the single-leg stance with eyes closed method are detailed in Table 4. The heterogeneity test revealed significant variation among the studies ($I^2$ = 86.574%, $p<0.001$), necessitating the use of a random-effects model for combining the effect sizes. The analysis confirmed that Taekwondo interventions significantly enhanced static balance ability (*ES* = 0.862, $p<0.001$).

*Sensitivity analysis.* To evaluate the robustness of the meta-analysis results for the single-leg stance with eyes closed method, a sensitivity analysis was performed using a sequential exclusion approach. This method involves excluding one study at a time and recalculating the overall effect size. The results, illustrated in Fig 4, indicate that the effect size range [0.696, 0.882] remains within the 95% confidence interval [0.487, 1.166] of the main effect meta-analysis, even after the exclusion of any single study. Furthermore, a visual inspection of the forest plot from the sensitivity analysis did not show any significant variations in effect sizes among the individual studies compared to the overall meta-analysis results. Consequently, these findings support the conclusion that the meta-analysis results are robust.

*Publication bias assessment.* Based on the funnel plot shown in Fig 5, asymmetry was observed, suggesting a potential publication bias. However, due to the limited number of studies included, direct assessment of publication bias using the funnel plot alone was not feasible.

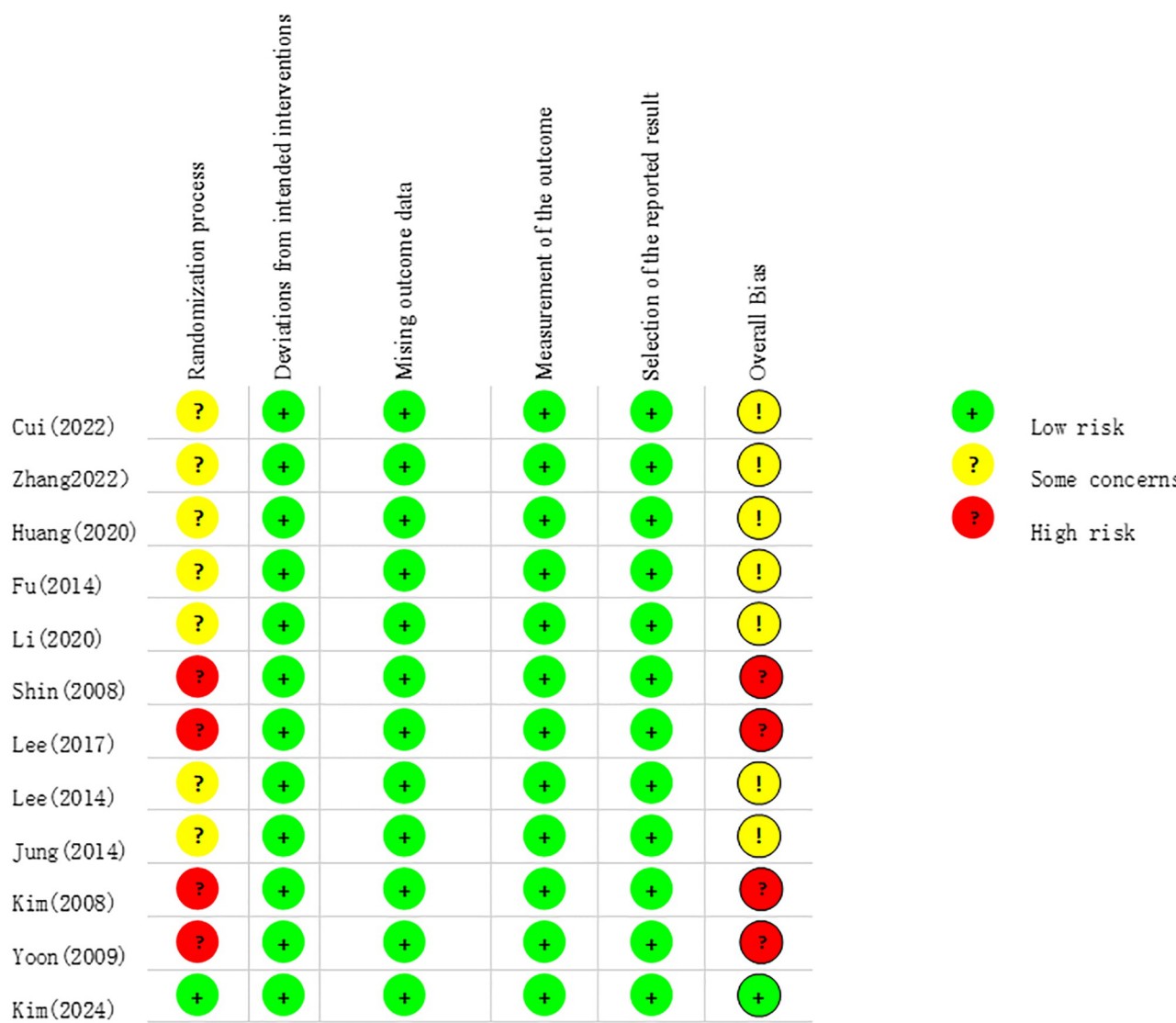

**Fig 3. Results of the RCTs risk of bias judgments about each risk of bias item for each included trail.**

Consequently, Egger's test was applied to more rigorously assess publication bias. The results of Egger's test yielded a p-value of 0.65, which exceeds the threshold of 0.05, indicating that there is no significant publication bias present in this meta-analysis.

*Subgroup analysis.* To investigate the sources of heterogeneity and determine the optimal intervention strategy for Taekwondo's impact on balance ability, subgroup analyses were conducted using Sex, Age, Period, Times per week, and minutes per times as moderator variables. A minimum of three studies is generally required for each subgroup to ensure representativeness [43]. To minimize the formation of invalid subgroups (those with fewer than three studies), similar subgroups were merged (e.g., merging the 4-6-year and 5-6-year subgroups into a single 4-6-year subgroup, and combining the 40-minute and 45-minute subgroups into a 40-45-minute subgroup). The final categorizations were as follows: Sex was divided into three subgroups: Unrestricted, Female, and Male; Age into three subgroups: 4–6, 9–13, and 18–21; Period into two subgroups: 12 weeks and 16 weeks; Times per week into two subgroups: once

**Table 3. Methodological quality assessment results of included before-and-after studies and cross-sectional studies.**

| The first author (Published year) | MINORS items | | | | | | | | Score |
|---|---|---|---|---|---|---|---|---|---|
| | 1 | 2 | 3 | 4 | 5 | 6 | 7 | 8 | |
| Chang (2020) | 2 | 2 | 2 | 2 | 0 | 0 | 0 | 1 | 9/16 |
| Shao (2023) | 2 | 2 | 2 | 2 | 0 | 0 | 0 | 1 | 9/16 |
| Yang (2014) | 2 | 2 | 2 | 2 | 0 | 0 | 0 | 1 | 9/16 |
| Wang (2018) | 2 | 2 | 2 | 2 | 0 | 0 | 0 | 1 | 9/16 |
| Pons Van Dijk (2013) | 2 | 2 | 1 | 2 | 0 | 0 | 0 | 1 | 8/16 |
| Pang (2016) | 2 | 2 | 2 | 2 | 0 | 0 | 0 | 1 | 9/16 |
| Guo (2023) | 2 | 2 | 2 | 2 | 0 | 0 | 0 | 1 | 9/16 |
| Pang (2015) | 2 | 2 | 2 | 2 | 0 | 0 | 0 | 1 | 9/16 |
| Chen (2016) | 2 | 2 | 2 | 2 | 0 | 0 | 0 | 1 | 9/16 |

Items:

1 = A clearly stated aim

2 = Inclusion of consecutive patients

3 = Prospective collection of data

4 = Endpoint appropriate to the aim of the study

5 = Unbiased assessment of the study endpoint

6 = Follow-up period appropriate to the aim of the study

7 = Loss to follow up less than 5%

8 = Prospective calculation of the study size

and 2–3 times; minutes per times into three subgroups: 40–45 minutes, 60–70 minutes, and 90–120 minutes. Subgroups that still had fewer than three studies after combining were excluded from the analysis.

The results of the subgroup analysis are detailed in Table 5. For the variable Sex, the intervention effect was greater in the Unrestricted group ($ES = 0.997$, $p = 0.001$) compared to the Female group ($ES = 0.786$, $p = 0.005$), while the effect in the Male group was not statistically significant ($p = 0.085$). Regarding Age, the intervention had a more pronounced effect in participants aged 18–21 years ($ES = 1.191$, $p = 0.001$) compared to those aged 4–6 years ($ES = 0.565$, $p = 0.003$); however, the effect for the 9–13 years group was not statistically significant ($p = 0.115$). For Period, the 12-week interventions ($ES = 1.375$, $p = 0.002$) demonstrated a greater effect than those spanning 16 weeks ($ES = 0.407$, $p = 0.003$). Concerning Times per week, interventions conducted once per week ($ES = 1.406$, $p = 0.003$) were more effective than those conducted 2–3 times per week ($ES = 0.434$, $p<0.001$). For minutes per time, timess lasting 60–70 minutes ($ES = 1.028$, $p<0.001$) showed a greater effect than those lasting 90–120 minutes ($ES = 0.969$, $p = 0.001$), while the effect of 40–45 minutes timess was not statistically significant ($p = 0.087$). Additionally, significant moderating effects were observed for Period, Times per week, and minutes per Time ($p<0.05$), but not for Sex and Age ($p>0.05$).

**Closed-eye marching in place.** In the literature reviewed, three studies evaluated dynamic balance using the closed-eye marching in place method. Given the limited number of studies,

**Table 4. Heterogeneity tests and quantitative synthesis results for single-leg stance with eyes closed.**

| Number of studies | Number of research subjects | ES (Hedges'g) | Test of null(2-Tail) | | 95%CI | | Heterogeneity test | |
|---|---|---|---|---|---|---|---|---|
| | | | z | p | lower limit | upper limit | $I^2$ | p |
| 14 | 774 | 0.862 | 4.775 | <0.001 | 0.487 | 1.166 | 86.574 | <0.001 |

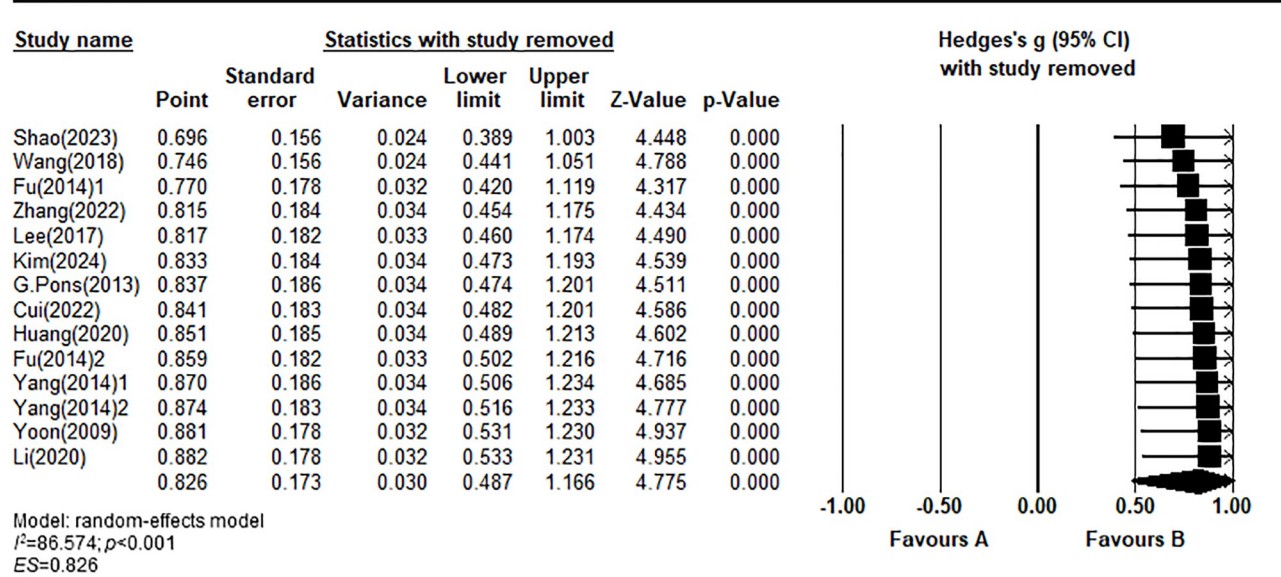

| Study name | | | | | | | |
|---|---|---|---|---|---|---|---|
| | | Statistics with study removed | | | | | |
| | Point | Standard error | Variance | Lower limit | Upper limit | Z-Value | p-Value |
| Shao(2023) | 0.696 | 0.156 | 0.024 | 0.389 | 1.003 | 4.448 | 0.000 |
| Wang(2018) | 0.746 | 0.156 | 0.024 | 0.441 | 1.051 | 4.788 | 0.000 |
| Fu(2014)1 | 0.770 | 0.178 | 0.032 | 0.420 | 1.119 | 4.317 | 0.000 |
| Zhang(2022) | 0.815 | 0.184 | 0.034 | 0.454 | 1.175 | 4.434 | 0.000 |
| Lee(2017) | 0.817 | 0.182 | 0.033 | 0.460 | 1.174 | 4.490 | 0.000 |
| Kim(2024) | 0.833 | 0.184 | 0.034 | 0.473 | 1.193 | 4.539 | 0.000 |
| G.Pons(2013) | 0.837 | 0.186 | 0.034 | 0.474 | 1.201 | 4.511 | 0.000 |
| Cui(2022) | 0.841 | 0.183 | 0.034 | 0.482 | 1.201 | 4.586 | 0.000 |
| Huang(2020) | 0.851 | 0.185 | 0.034 | 0.489 | 1.213 | 4.602 | 0.000 |
| Fu(2014)2 | 0.859 | 0.182 | 0.033 | 0.502 | 1.216 | 4.716 | 0.000 |
| Yang(2014)1 | 0.870 | 0.186 | 0.034 | 0.506 | 1.234 | 4.685 | 0.000 |
| Yang(2014)2 | 0.874 | 0.183 | 0.034 | 0.516 | 1.233 | 4.777 | 0.000 |
| Yoon(2009) | 0.881 | 0.178 | 0.032 | 0.531 | 1.230 | 4.937 | 0.000 |
| Li(2020) | 0.882 | 0.178 | 0.032 | 0.533 | 1.231 | 4.955 | 0.000 |
| | 0.826 | 0.173 | 0.030 | 0.487 | 1.166 | 4.775 | 0.000 |

Model: random-effects model
$I^2=86.574; p<0.001$
ES=0.826

**Meta Analysis**

**Fig 4. Forest plot of sensitivity analysis for single-leg stance with eyes closed.**

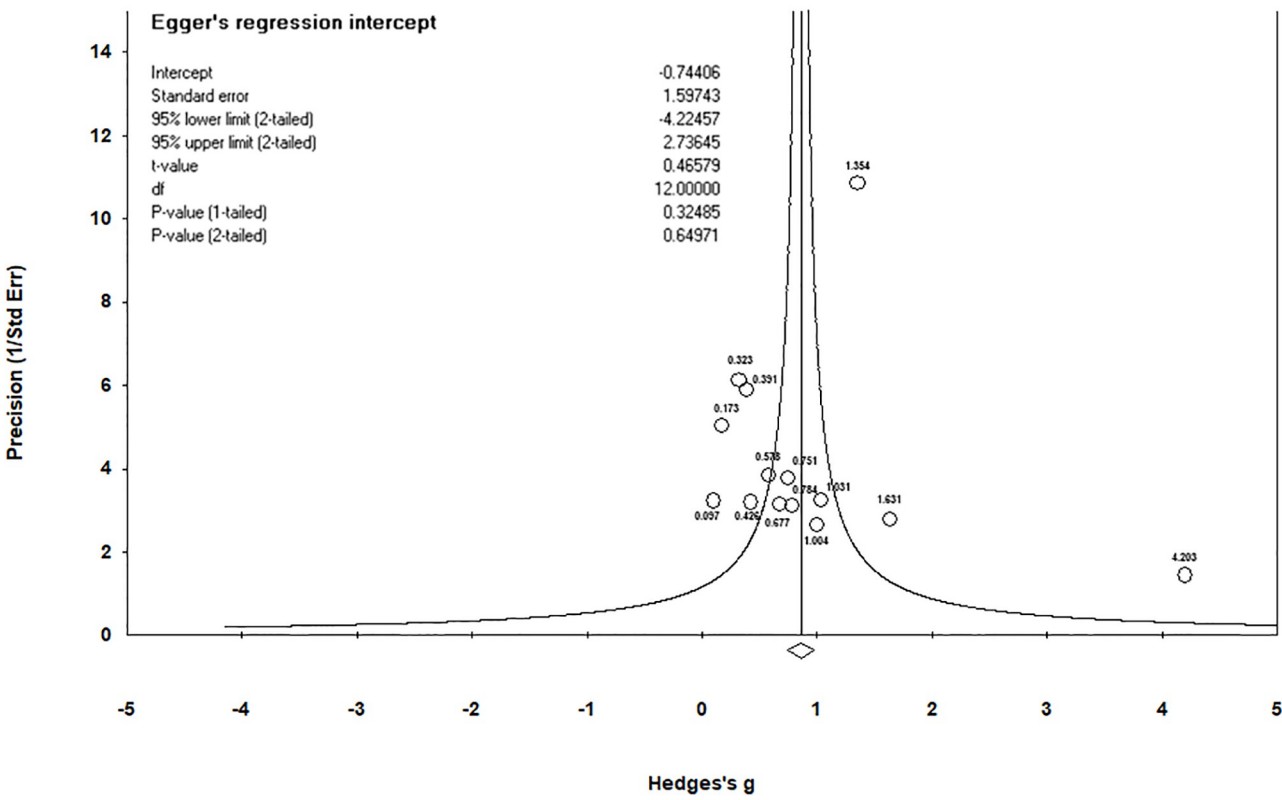

**Fig 5. Funnel plot.**

**Table 5. Results of subgroup analysis.**

| Moderating variable | Subgroup | Number of articles | Moderator (p) | Heterogeneity test | | ES | Test of null(2-Tail) | |
|---|---|---|---|---|---|---|---|---|
| | | | | $I^2$ | p | | z | p |
| Sex | Unrestricted | 6 | 0.786 | 85.031 | <0.001 | 0.997 | 3.183 | 0.001 |
| | Female | 5 | | 88.685 | <0.001 | 0.786 | 2.804 | 0.005 |
| | Male | 3 | | 83.323 | 0.002 | 0.669 | 1.724 | 0.085 |
| Age | 4–6 | 4 | 0.300 | 50.523 | 0.109 | 0.565 | 2.984 | 0.003 |
| | 9–13 | 3 | | 82.036 | 0.004 | 0.702 | 1.575 | 0.115 |
| | 18–21 | 5 | | 93.928 | <0.001 | 1.191 | 3.342 | 0.001 |
| | 41–70 | 1 | | - - - | | | | |
| | 65–75 | 1 | | - - - | | | | |
| Period | 12week | 5 | 0.037 | 86.179 | <0.001 | 1.375 | 3.098 | 0.002 |
| | 16week | 4 | | 46.930 | 0.130 | 0.407 | 2.976 | 0.003 |
| | 4 months | 1 | | - - - | | | | |
| | 8week | 1 | | - - - | | | | |
| | 10week | 1 | | - - - | | | | |
| | 15week | 1 | | - - - | | | | |
| | 15 months | 1 | | - - - | | | | |
| Times per week | 1 | 4 | 0.045 | 87.686 | <0.001 | 1.406 | 2.969 | 0.003 |
| | 2–3 | 6 | | 19.860 | 0.284 | 0.434 | 4.162 | <0.001 |
| | 5 | 2 | | - - - | | | | |
| | not reported | 2 | | - - - | | | | |
| Minutes per time | 40-50min | 3 | 0.009 | 8.149 | 0.337 | 0.269 | 1.714 | 0.087 |
| | 60-70min | 3 | | 71.146 | 0.031 | 1.028 | 4.331 | <0.001 |
| | 90-120min | 6 | | 85.298 | <0.001 | 0.969 | 3.438 | 0.001 |
| | not reported | 2 | | - - - | | | | |

"- - -": Invalid subgroup

additional analyses for publication bias and sensitivity were considered unnecessary, resulting in the discontinuation of more detailed analysis for this method. The heterogeneity tests and meta-analysis results are presented in Table 6. These tests showed no significant heterogeneity ($I^2 = 55.927$, $p = 0.103$), leading to the adoption of a fixed-effects model for combining effect sizes. The results demonstrated that Taekwondo interventions significantly enhance dynamic balance ability ($ES = 0.296$, $p = 0.036$).

**Win pod.** Within the selected literature, three cross-sectional studies employed the Win pod device to assess static balance ability. One study classified participants according to obesity status [20], while another differentiated participant into age groups of 6–8 years and 9–12 years [29]. As a result, data were aggregated into five groups (treated as five individual studies) for meta-analysis purposes.

The results of the heterogeneity tests and the meta-analysis for the Win pod assessments are presented in Table 7. The analysis revealed substantial heterogeneity for the "X speed (mm/s)"

**Table 6. Heterogeneity tests and meta-analysis results for marching in place with eyes closed.**

| Number of studies | Number of research subjects | ES (Hedges'g) | Test of null(2-Tail) | | 95%CI | | Heterogeneity test | |
|---|---|---|---|---|---|---|---|---|
| | | | z | p | lower limit | upper limit | $I^2$ | p |
| 3 | 116 | 0.296 | 2.093 | 0.036 | 0.019 | 0.573 | 55.927 | 0.103 |

Table 7. Heterogeneity tests and meta-analysis results for Win pod.

| Number of studies | Number of research subjects | Indicators | ES (Hedges'g) | Test of null (2-Tail) | | 95%CI | | Heterogeneity test | |
|---|---|---|---|---|---|---|---|---|---|
| | | | | $z$ | $p$ | lower limit | upper limit | $I^2$ | $p$ |
| 5 | 140 | Length(mm) | 0.281 | 1.674 | 0.094 | -0.048 | 0.610 | 57.868 | 0.050 |
| | | Area(mm$^2$) | -0.020 | -0.122 | 0.903 | -0.343 | 0.303 | 0.000 | 0.619 |
| | | Avg.V(mm/s) | 0.277 | 1.654 | 0.098 | -0.051 | 0.606 | 56.024 | 0.059 |
| | | X speed(mm/s) | 0.443 | 1.542 | 0.123 | -0.120 | 1.005 | 64.817 | 0.023 |
| | | Y speed(mm/s) | 0.160 | 0.957 | 0.339 | -0.168 | 0.487 | 54.476 | 0.067 |
| | | X dev.(mm) | 0.280 | 1.672 | 0.094 | -0.048 | 0.609 | 56.527 | 0.056 |
| | | Y dev.(mm) | 0.108 | 0.650 | 0.515 | -0.216 | 0.432 | 2.649 | 0.391 |

variable ($I^2$ = 64.817, $p$ = 0.023), prompting the use of a random-effects model for this specific parameter, whereas a fixed-effects model was applied to other parameters. The meta-analysis indicated that none of the indicators measured by the Win pod reached statistical significance ($p>0.05$), suggesting that Taekwondo interventions do not significantly affect static balance as assessed by the Win pod.

## Qualitative analysis

Due to the infrequent use of certain balance ability measurement methods, which appeared only once or twice among the included studies, conducting a meta-analysis was not feasible. Consequently, a qualitative analysis was undertaken to more thoroughly explore the potential impact of Taekwondo training on balance ability. This analysis involved an in-depth interpretation of the studies, which included randomized controlled trials, before-and-after studies, and cross-sectional studies. These studies featured participants ranging from young children to elderly women, with intervention durations varying from 6 weeks to 15 months.

For the qualitative analysis, in addition to the studies included in the quantitative synthesis (employing single-leg stance with eyes closed, marching in place with eyes closed, and Win pod), additional studies that used the "Force Plate" and "Balance Performance Monitor" for measuring static balance ability were also reviewed. Furthermore, the analysis incorporated studies that applied various methods and equipment to assess dynamic balance ability, including "walking on a balance beam," "Y balance test," "star excursion balance test," "single-leg jump," "timed up-and-go test," "functional reach test," "Good Balance System," and "GAITRite."

**Static balance ability.** Shin (2008) assessed static balance ability using a "Force Plate," observing significant improvements in participants after a 12-week Taekwondo intervention. Specifically, enhancements were noted in the center of pressure (COP) trajectory across both anterior-posterior and medial-lateral dimensions, including range, distance, velocity, mean velocity, and the area of the 95% confidence ellipse ($p<0.05$). Kim (2008) utilized the "Balance Performance Monitor" to evaluate static balance, focusing on metrics such as "Max velocity," "Sway path," and "Sway area." Following a 6-week Taekwondo intervention, there were significant reductions in both "Max velocity" and "Sway path" ($p<0.05$), although no significant changes were noted in the "Sway area" ($p>0.05$).

**Dynamic balance ability.** This review consolidates multiple studies assessing the impact of Taekwondo on dynamic balance using diverse evaluation methods. Cui (2022) and Guo

(2023) applied "balance beam walking" to assess the dynamic balance of children aged 4–6. Their findings indicated that, following a 12-week intervention, the Taekwondo group significantly outperformed the control group in balance beam walking duration ($p < 0.05$).

Lee (2017) and Guo (2023) employed the "Y-balance test" to quantify dynamic balance by measuring lower limb extension. Both studies reported significant improvements in extension values for Taekwondo groups among 4-6-year-olds and female university students ($p < 0.05$).

Pang (2016) and Chen (2016) utilized the "star excursion test" to assess dynamic balance through the distance extended by both feet in three directions: anterior, posterior medial, and posterior lateral. Pang (2016) observed no significant improvements in the left foot's posterior medial and right foot's anterior metrics compared to controls ($p > 0.05$), whereas all other metrics demonstrated significant enhancements ($p < 0.05$). Conversely, Chen (2016) documented significant improvements in all metrics for the Taekwondo group ($p < 0.05$).

Pons Van Dijk (2013) measured dynamic balance using the "single-leg jump" for individuals over 40, finding no significant changes post-intervention ($p > 0.05$).

Kim (2024) assessed dynamic balance in elderly women using the "timed up-and-go test (TUGT)" and "functional reach test (FRT)," noting significant post-intervention improvements in both TUGT performance and reach distance ($p < 0.05$).

Lee (2014) and Jung (2014) evaluated dynamic balance with the "Good Balance System," measuring anterior-posterior, medial-lateral, and total center of pressure (COP) distances. They found significant improvements in all metrics for the Taekwondo groups compared to controls ($p < 0.05$).

Lastly, Kim (2008) analyzed dynamic balance in older adults using the "GAITRite" system, observing significant enhancements in most assessed metrics after six weeks of Taekwondo training ($p < 0.05$), except for specific measures such as "Step Time" and "Single Stance Time."

These studies collectively underscore the positive effects of Taekwondo training on dynamic balance across various age groups and assessment methods, suggesting its utility in enhancing motor control and stability.

In summary, while certain studies displayed no significant changes in specific metrics, the overarching results affirm that Taekwondo training positively influences both static and dynamic balance abilities. The majority of the research indicates that Taekwondo interventions notably enhance static balance, especially in terms of center of pressure (COP) control. Particularly, studies employing the "Force Plate" and "Balance Performance Monitor" have identified multiple significant improvements across various static balance indicators, highlighting Taekwondo's effectiveness in bolstering postural control capabilities. For dynamic balance, the findings from various assessment methods consistently demonstrate significant enhancements in lower limb extension capability and walking stability. For instance, studies using "balance beam walking" and the "Y-balance test" reported that participants in the Taekwondo intervention groups significantly outperformed their control counterparts, underlining the efficacy of Taekwondo in improving dynamic balance. Moreover, applications of the "timed up-and-go test (TUGT)" and "functional reach test (FRT)" have shown that Taekwondo can substantially enhance dynamic balance in elderly women, as evidenced by significant reductions in TUGT times and increased reach distances following a 12-week training program. Although these qualitative analyses do not provide statistical conclusions of significance, they offer valuable insights into the various ways Taekwondo training influences balance abilities. Collectively, these results support and augment the quantitative findings, further validating the positive impacts of Taekwondo training across different demographics and assessment techniques.

## Discussion

### Sources of heterogeneity and moderating effects in closed-eye one-leg standing

Due to data constraints, subgroup analysis was limited to the single-leg stance with eyes closed method, which confined the exploration of heterogeneity and moderating effects to this specific assessment. The study addressed the sources of statistical heterogeneity by evaluating differences among subgroups. With significant heterogeneity observed in the main effect meta-analysis ($I^2$ = 86.574%, $p$<0.001), a potential moderator variable would be considered a source of statistical heterogeneity if it consistently showed non-significant heterogeneity across all subgroups. However, as indicated in Table 5, none of the moderator variables displayed non-significant heterogeneity in all their subgroups, which hindered the identification of a definitive source of statistical heterogeneity. Additionally, two studies employing the single-leg stance with eyes closed method were noted for having a high risk of bias [19, 41], which could contribute to methodological heterogeneity. Nonetheless, sensitivity analysis demonstrated that the exclusion of any single study did not significantly alter the effect size, underscoring the robustness and reliability of the meta-analysis results. The study also examined the moderating effects of Sex, Age, Period, Times per week, and minutes per Times on the efficacy of Taekwondo interventions on static balance ability. The results indicated that Period, Times per week, and minutes per Time significantly influenced the intervention effect ($p$<0.05), suggesting these factors are critical in optimizing the efficacy of Taekwondo training. In contrast, Sex and Age did not significantly impact the intervention effect ($p$>0.05), highlighting that the benefits of Taekwondo on static balance are consistent across different genders and age groups.

### Main effects

This study assessed the impact of Taekwondo training on balance ability using both quantitative and qualitative analyses. While some indicators did not show significant results, overall, the findings indicate that Taekwondo training positively affects both static and dynamic balance abilities. Specifically, quantitative analysis within the meta-analysis highlighted significant effects from exercises such as single-leg stance and marching in place with eyes closed. Qualitative analysis further revealed notable improvements in postural control, lower limb extension, and gait stability among participants engaged in Taekwondo. This aligns with previous meta-analyses which have demonstrated that other forms of exercise, including aerobic exercise [44], Tai Chi [45], and Ba Duan Jin [7], similarly enhance balance ability. Moreover, previous studies have found that training in combat sports such as judo [46], karate [47], and boxing [48] has been shown to positively impact balance abilities. These findings align with the results of the present study and indirectly support the effectiveness of Taekwondo in enhancing balance. Taekwondo training incorporates various posture-maintaining exercises that involve standing stances and leg control techniques, alongside fast-paced movements, jumps, and kicks. These activities are instrumental in strengthening lower limb muscles, enhancing endurance, and developing explosive power, which are critical for improving both static and dynamic balance. Furthermore, research has indicated that Taekwondo interventions can enhance attention [49], a key component necessary for maintaining balance.

The study also uncovered some negative results. Quantitative analysis showed a non-significant combined effect ($p$>0.05) for cross-sectional studies that used the Win pod to assess static balance ability. It is crucial to acknowledge, however, that this combined result derived from only three articles, which may not adequately represent broader trends. Furthermore, the

small sample size could have diminished statistical power, thus increasing the likelihood of type II errors—failing to detect an actual effect [50]. From another angle, the quantitative synthesis of this study primarily utilized the "single-leg stance," a method that depends on observation and timing to assess balance. While this method is straightforward and convenient, it is prone to human error and subjective interpretation. Although commonly used, it may not capture subtle changes as effectively as more objective tools. Objective equipment can provide detailed quantitative data such as center of pressure (COP) trajectories, velocity, and acceleration, offering a more accurate and objective measure of balance ability [51]. This is particularly crucial for detecting minute changes that might elude visual observation or timing techniques. Therefore, future studies should employ objective equipment to measure balance ability, enhancing the objectivity and reliability of the findings.

## Subgroup analysis of closed-eye one-leg standing

Subgroup analysis was conducted to explore gender differences in the impact of Taekwondo on static balance ability. Results indicated a significant improvement in static balance among females ($ES = 0.768$, $p = 0.005$), while the effect in males was not statistically significant ($p = 0.085$). Although the intervention effect was greatest in the unrestricted group ($ES = 0.997$, $p = 0.001$), the data suggest that females may benefit more from Taekwondo in enhancing static balance compared to males. Typically, male athletic performance is considered superior to that of females [52, 53]. however, these results suggest otherwise. In the studies analyzed, both male and female participants engaged in exercises at equal (moderate) intensity, which might explain the greater improvements observed in females under identical conditions. Additionally, other research supports that females often experience greater benefits from exercise, particularly in terms of lower limb strength [54, 55], which is consistent with our findings. This gender disparity warrants further attention, and future studies should carefully consider gender variations in their research design.

Subgroup analysis further examined age-related differences in the impact of Taekwondo on static balance ability. The largest intervention effect was observed in the 18-21-year-old group ($ES = 1.191$, $p = 0.001$), followed by the 4-6-year-old group ($ES = 0.565$, $p = 0.003$). The 9-13-year-old group, however, showed no significant effect ($p > 0.05$). These findings suggest that Taekwondo might exert a more substantial intervention effect on balance ability in adults than in minors. This discrepancy could be due to the complexity of certain Taekwondo maneuvers, which might be too challenging for younger participants to perform effectively, potentially leading to suboptimal training outcomes. Physiologically, adults have fully developed muscular and nervous systems, which provide enhanced muscle strength and neural control, contributing to superior exercise performance. Conversely, minors, whose neuromuscular systems are still developing, may face greater difficulties in coordinating and controlling complex movements.

In this study, we explored the variations in the effects of Taekwondo interventions on static balance ability under different intervention durations to identify the optimal program length. Subgroup analysis revealed that a 12-week intervention ($ES = 1.375$, $p = 0.002$) was more effective than a 16-week intervention ($ES = 0.407$, $p = 0.003$), suggesting that longer intervention periods do not necessarily result in better outcomes. This finding aligns with previous research, which indicated that interventions of less than 10 weeks often had greater positive effects on balance compared to those extending beyond 12 weeks [56]. Another study demonstrated that an 8- to 12-week jump rope training program was more beneficial for physical fitness, including balance, than longer programs [57]. It is important to note that these findings do not imply that longer interventions are ineffective for improving balance; rather, the

effectiveness of an intervention likely depends on a combination of factors including the frequency of times, duration per times, and overall intensity. Given the limitations in the scope of current research, a more detailed subgroup analysis was not feasible. Future studies should therefore include a broader range of intervention durations to gain a more comprehensive understanding of their impacts. However, based on the available evidence, a 12-week Taekwondo program appears to be most effective for enhancing static balance ability.

"Times per week" and "minutes per time" are critical variables that together define the total weekly intervention duration in Taekwondo training. Subgroup analysis of "Times per week" revealed that training once per week ($ES = 1.406$, $p = 0.003$) was more effective than training 2–3 times per week ($ES = 0.434$, $p<0.001$). This indicates that a higher frequency of training does not necessarily enhance the intervention effect, challenging assumptions supported by previous studies [58, 59]. Unlike general physical exercise, Taekwondo often involves training content with a certain level of difficulty. The participants in this study were beginners with no prior experience in Taekwondo. Excessively high training frequencies could potentially increase their psychological stress, which might, in turn, affect the effectiveness of the intervention. Additionally, Previous studies on exercise interventions have also identified a non-linear relationship between training frequency and exercise effectiveness. For instance, a community-based exercise study demonstrated that training twice per week was more beneficial than three times per week [60]. Furthermore, a study assessing exercise interventions in older adults indicated that training three times per week was more effective than four times per week [61], highlighting that excessive training frequency might lead to overtraining syndrome, adversely affecting performance and outcomes [62]. In terms of "minutes per time," subgroup analysis showed that timess lasting 60–70 minutes had the highest intervention effect ($ES = 1.028$, $p<0.001$), followed by timess of 90–120 minutes ($ES = 0.969$, $p = 0.001$), while timess lasting 40–50 minutes showed no significant effect ($p = 0.087$). This suggests a potential inverted U-shaped relationship between minutes per time and intervention effectiveness, echoing findings from earlier studies [60, 63]. To optimize intervention outcomes, a balanced approach to times per week and minutes per time is necessary. Current evidence supports the notion that a weekly Taekwondo times lasting 60–70 minutes might be the most effective strategy for improving static balance ability.

## Limitations of the study

This study utilized both quantitative and qualitative assessments to evaluate the impact of Taekwondo on balance ability, yet it is important to acknowledge several limitations. Firstly, the scope of the quantitative synthesis was restricted to three types of balance assessments: "single-leg stance with eyes closed," "marching in place with eyes closed," and "Win pod." These methods cover both static and dynamic balance, yet the representation of dynamic balance, particularly through "marching in place with eyes closed," was limited. Furthermore, the methods "single-leg stance" and "marching in place with eyes closed" are prone to human error, necessitating careful interpretation of the results. Secondly, the analysis did not adequately address exercise intensity due to insufficient data, precluding a more detailed subgroup analysis. Thirdly, to include as many studies as possible in the subgroup analysis, similar subgroups were merged, which might have introduced discrepancies in the final results. Fourthly, even after merging some subgroups, there remained a significant number of invalid subgroups, which constrained the breadth of the final analysis. Fifthly, although sensitivity analysis and subgroup analysis were employed to pinpoint the sources of heterogeneity, their specific origins could not be conclusively determined. In summary, while the qualitative and quantitative

analyses have identified certain trends, these findings should be approached with caution given the study's various limitations.

## Conclusion

This study confirms that Taekwondo training can effectively enhance both static and dynamic balance abilities, with indications of particularly significant benefits for static balance in adult females. Due to data constraints, subgroup analysis was confined to static balance ability, specifically using the single-leg stance with eyes closed method. Based on the evidence available, it is recommended that a 12-week Taekwondo training regimen, consisting of one 60–70 minutes times per week, be implemented to optimize the impact on static balance ability. However, given the limitations of this study, there is a need for further high-quality research that delves deeper into these findings to confirm and elaborate on the results.

## Supporting information

**S1 Data. List of retrieved studies and original research data.**
(XLS)

**S2 Data. Relevant signaling questions and answers.**
(XLS)

**S1 File. Search strategy.**
(DOCX)

**S2 File. Description of balance assessment methods.**
(DOCX)

**S1 Checklist. PRISMA 2020 checklist.**
(DOCX)

## Author Contributions

**Conceptualization:** Hanyu Ju.

**Data curation:** Zhengfa Han.

**Formal analysis:** Zhengfa Han.

**Funding acquisition:** Hanyu Ju.

**Methodology:** Zhengfa Han, Hanyu Ju.

**Writing – original draft:** Zhengfa Han.

**Writing – review & editing:** Hanyu Ju.

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
