## [Decision Letter · Decision Letter 0]

9 Dec 2024

PONE-D-24-37293Effects of Taekwondo Intervention on Balance Ability: A Meta-Analysis and Systematic ReviewPLOS ONE

Dear Dr. Ju,

Thank you for submitting your manuscript to PLOS ONE. After careful consideration, we feel that it has merit but does not fully meet PLOS ONE’s publication criteria as it currently stands. Therefore, we invite you to submit a revised version of the manuscript that addresses the points raised during the review process.

We look forward to receiving your revised manuscript.

Kind regards,

Artur Kruszewski

Academic Editor

PLOS ONE

Journal Requirements:

“This study complies with the current laws of the country/region where it was conducted, and there are no conflicts of interest. The study was funded by the Talent Introduction Project of Sanya University (USYRC24-07).”

3. Please ensure that you refer to Figure 5 in your text as, if accepted, production will need this reference to link the reader to the figure. 

4. Please upload a new copy of Figure 3 as the detail is not clear. Please follow the link for more information: 

https://blogs.plos.org/plos/2019/06/looking-good-tips-for-creating-your-plos-figures-graphics/

https://blogs.plos.org/plos/2019/06/looking-good-tips-for-creating-your-plos-figures-graphics/

5. We note you have included a table to which you do not refer in the text of your manuscript. Please ensure that you refer to Tables 6 and 7 in your text; if accepted, production will need this reference to link the reader to the Table.

7. As required by our policy on Data Availability, please ensure your manuscript or supplementary information includes the following: 

Reviewers' comments:

Reviewer's Responses to Questions

**Comments to the Author**

1. Is the manuscript technically sound, and do the data support the conclusions?

Reviewer #1: Yes

Reviewer #2: Yes

2. Has the statistical analysis been performed appropriately and rigorously? 

Reviewer #1: Yes

Reviewer #2: Yes

3. Have the authors made all data underlying the findings in their manuscript fully available?

Reviewer #1: Yes

Reviewer #2: Yes

4. Is the manuscript presented in an intelligible fashion and written in standard English?

Reviewer #1: Yes

Reviewer #2: Yes

5. Review Comments to the Author

Reviewer #1: Dear Authors,

thank you for the opportunity to perform a review of your article “Effects of Taekwondo Intervention on Balance Ability: A Meta-Analysis and Systematic Review”. The article is correctly divided into sections. The article is written according to current PRISMA standards for review. The authors performed an objective review in leading science databases. But I suggest two corrections that, in my opinion, will contribute to a better perception of your work.

1. Change the keywords to something other than in the title - this will allow your work to be better identified in the databases.

2. Expand the discussion to include a reference of the established state of knowledge about balance in Taekwondo to several works from other combat sports, karate, judo, etc. By comparison, this will highlight the significant size of your review and contribution to the current state of knowledge of Taekwondo.

In tek, consider referring to: Maslinski J et al. Original methods and tools used for studies on the body balance disturbance tolerance skills of the Polish judo athletes from 1976 to 2016. Arch Budo 2017; 13: 285-296.

Reviewer #2: Review for research paper "Effects of Taekwondo Intervention on Balance Ability: A Meta-Analysis and Systematic Review" by Han Zhengfa and Ju Hanyu

Dear Editor,

The paper "Effects of Taekwondo Intervention on Balance Ability: A Meta-Analysis and Systematic Review" presents a comprehensive meta-analysis and systematic review of the effects of Taekwondo interventions on balance ability. The study is well-structured, with a clear objective, detailed methodology, and thorough quantitative and qualitative data analysis. The findings indicate that the data supports the proposed optimal intervention protocol of 60-70 minutes per session, once weekly for 12 weeks.

The paper is well-structured and comprehensive, but a few areas could be improved. I recommend publishing the manuscript "Effects of Taekwondo Intervention on Balance Ability: A Meta-Analysis and Systematic Review" after a major review. The following issues should be improved or clarified (the pages were numbered starting from the manuscript's page 9):

1. P 9. Line 27: To add the results of bias for studies. Was it low or high?

2. P 10. L 33. What kind of training do you mean in this sentence: for and a times length of 60-70 minutes? Taekwondo or balance training?

3. P 12. L 60. What kind of intervention? Explain in more detail.

4. P 12. L 73. Specify what PRISMA stands for.

5. P 13. L 79. Add the starting date as well.

6. P 13. L 90. Write the full name for PICOS.

7. P 13. L 94. My suggestion is, instead of Taekwondo Gymnastic, to write Freestyle taekwondo.

8. P 14. L 106. Which version? Write a proper citation.

9. P 14. L 113. Reference it.

10. P 15. L 129 and 130 are redundant and previously repeated. Delete them.

11. P 16. L 157. Explain briefly the moderate intensity of 7 studies. Were they based on heart rate or other factors?

12. P 17. L 159. Explain the test "single-leg stance with eyes closed."

13. P 18 and 19. In Table 1, the Period section specifies whether it is a Week or a Month.

14. P 19. For Pons Van Dijk's (2013) reference, it is unclear whether there are 40 sessions per month or week.

15. P 19. Correct the size of 60 min in Table 1 for Kim (2024).

16. P 21. Do you mean there are in this sentence? I think it’s a typo mistake. Regarding overall bias, the ame four studies.

17. P21. Write RTCs instead of rcts for both Fig. 2 and Fig. 3.

18. P 24. Please write the statistical analysis summary in Figure 4.

19. P 24. In this sentence, Visual inspection of the funnel plot, you should write (Figure 5).

20. P 24. Where is the result of Egger's test?

21. P 24. Having the effect sizes per study name inside the funnel plot in Fig. 5 will be great—funnel Plot.

22. P 26. There is a misinterpretation of the results and p-value for Table 5. In the last paragraph of this page, you wrote: Additionally, significant moderating effects were observed for Period, Times per week, and minutes per Time (p<0.05), but not for Sex and Age (p>0.05). The p-values for gender in male and female groups and Ages 4-6 and 18-21 are significant.

23. P 27. You should change and write Table 6 in the text instead of Table 4 in this sentence: for this method. The heterogeneity tests and meta-analysis results are presented in Table 4.

24. P 27. All tables indicate the significant value for p. Is it p<0.05 or p<0.01?

25. P 28. You should change and write Table 7 in the text instead of Table 5 in this sentence: presented in Table 5. The analysis revealed substantial heterogeneity for the "X speed (mm/s)."

26. Table 7 shows the heterogeneity results for length (mm), which show I2 = 57.868 and p =0.050. Isn't heterogeneity marginally significant?

27. P 33. L 102 – 106, revise based on 22—p 26.

28. P 37. L 188. Which study? Write the reference number.

29. P 48. For Fig. 4, the overall estimated effect size and heterogeneity test results are needed.

6. PLOS authors have the option to publish the peer review history of their article (what does this mean?). If published, this will include your full peer review and any attached files.

Reviewer #1: No

Reviewer #2: No

---

## [Author Response · Author response to Decision Letter 0]

30 Dec 2024

We have addressed the comments and suggestions from the academic editor and reviewers and revised our manuscript accordingly. Additionally, we have added some supplementary materials. Please refer to the "Supplementary Files" for details.

---

## [Editor Report · Decision Letter 1]

7 Jan 2025

Effects of Taekwondo Intervention on Balance Ability: A Meta-Analysis and Systematic Review

PONE-D-24-37293R1

Dear Dr Hanyu Ju. 

We’re pleased to inform you that your manuscript has been judged scientifically suitable for publication and will be formally accepted for publication once it meets all outstanding technical requirements.

Kind regards,

Artur Kruszewski

Academic Editor

PLOS ONE
---

## [Editor Report · Acceptance letter]

14 Jan 2025

PONE-D-24-37293R1 

PLOS ONE

Dear Dr. Ju, 

I'm pleased to inform you that your manuscript has been deemed suitable for publication in PLOS ONE. Congratulations! Your manuscript is now being handed over to our production team.

Kind regards, 

on behalf of

PhD Artur Kruszewski 

Academic Editor

PLOS ONE